# Reconstruction of a catalogue of genome-scale metabolic models with enzymatic constraints using GECKO 2.0

Iván Domenzain [1,2], Benjamín Sánchez [3,4,9], Mihail Anton [1,5,9], Eduard J. Kerkhoven [1,2], Aarón Millán-Oropeza[6], Céline Henry [6], Verena Siewers [1,2], John P. Morrissey [7], Nikolaus Sonnenschein[3] & Jens Nielsen [1,2,8 ✉]

Genome-scale metabolic models (GEMs) have been widely used for quantitative exploration of the relation between genotype and phenotype. Streamlined integration of enzyme constraints and proteomics data into such models was first enabled by the GECKO toolbox, allowing the study of phenotypes constrained by protein limitations. Here, we upgrade the toolbox in order to enhance models with enzyme and proteomics constraints for any organism with a compatible GEM reconstruction. With this, enzyme-constrained models for the budding yeasts *Saccharomyces cerevisiae*, *Yarrowia lipolytica* and *Kluyveromyces marxianus* are generated to study their long-term adaptation to several stress factors by incorporation of proteomics data. Predictions reveal that upregulation and high saturation of enzymes in amino acid metabolism are common across organisms and conditions, suggesting the relevance of metabolic robustness in contrast to optimal protein utilization as a cellular objective for microbial growth under stress and nutrient-limited conditions. The functionality of GECKO is expanded with an automated framework for continuous and version-controlled update of enzyme-constrained GEMs, also producing such models for *Escherichia coli* and *Homo sapiens*. In this work, we facilitate the utilization of enzyme-constrained GEMs in basic science, metabolic engineering and synthetic biology purposes.

[1] Department of Biology and Biological Engineering, Chalmers University of Technology, SE-412 96 Gothenburg, Sweden. [2] Novo Nordisk Foundation Center for Biosustainability, Chalmers University of Technology, SE-412 96 Gothenburg, Sweden. [3] Department of Biotechnology and Biomedicine, Technical University of Denmark, 2800 Kongens Lyngby, Denmark. [4] Novo Nordisk Foundation Center for Biosustainability, Technical University of Denmark, 2800 Kongens Lyngby, Denmark. [5] Department of Biology and Biological Engineering, National Bioinformatics Infrastructure Sweden, Science for Life Laboratory, Chalmers University of Technology, Kemivägen 10, SE-412 58 Gothenburg, Sweden. [6] Plateforme d'analyse protéomique Paris Sud-Ouest (PAPPSO), INRAE, MICALIS Institute, Université Paris-Saclay, 78350 Jouy-en-Josas, France. [7] School of Microbiology, Environmental Research Institute and APC Microbiome Ireland, University College Cork, T12 K8AF Cork, Ireland. [8] BioInnovation Institute, Ole Maaløes Vej 3, 2200 Copenhagen, Denmark. [9]These authors contributed equally: Benjamín Sánchez, Mihail Anton. ✉email: nielsenj@chalmers.se

Genome-scale metabolic models (GEMs) have become an established tool for systematic analyses of metabolism for a wide variety of organisms[1–6]. Their myriads of applications span from model-driven development of efficient cell factories[3,7–9], to their utilization for understanding mechanisms underlying complex human diseases[10–12]. One of the most common simulation techniques for enabling phenotype predictions with these models is flux balance analysis (FBA), which assumes that there is balancing of fluxes around each metabolite in the metabolic network. This means that fluxes are constrained by stoichiometries of the biochemical reactions in the network, and that cells have evolved in order to operate their metabolism according to optimality principles[13,14]. Quantitative determination of biologically meaningful flux distribution profiles is a major challenge for constraint-based methods, as optimal phenotypes can be attained by alternate flux distribution profiles[15], caused by the presence of network redundancies that provide organisms with robustness to environmental and genetic perturbations. This limitation is often addressed by incorporation of experimental measurements of exchange fluxes (secretion of byproducts and uptake of substrates) as numerical flux constraints for the FBA problem. However, such measurements are not readily available for a wide variety of conditions and organisms.

In order to overcome these limitations, the concept of enzymatic limitations on metabolic reactions has been explored and incorporated by several constraint-based methods. Some of these have modeled enzyme demands of metabolic reactions by constraining metabolic networks with kinetic parameters and physiological limitations of cells, such as a crowded intracellular volume[16–18], a finite membrane surface area for expression of transporter proteins[19] and a bounded total protein mass available for metabolic enzymes[20–25]. All of these modeling frameworks have been successful at expanding the range of predictions of classical FBA, providing explanations for overflow metabolism and cellular growth on diverse environments for *Escherichia coli*[16–19,21,23,25], *Saccharomyces cerevisiae*[22,25,26], *Lactococus lactis*[27], and even human cells[20,24]. However, these modeling approaches were applied to metabolic networks of extensively studied model organisms, which are usually well represented in specialized resources for kinetic parameters such as the BRENDA[28] and SABIO RK[29] databases. Furthermore, collecting the necessary parameters for the aforementioned models was mostly done manually; therefore, no generalized model parameterization procedure was provided as an integral part of these methods.

Enzyme limitations have also been introduced into models of metabolism by other formalisms, for instance, Metabolic and gene Expression models (ME-models), implemented on reconstructions for *E. coli*[30–33], *Thermotoga maritima*[34] and *Lactococus lactis*[35]; and resource balance analysis models (RBA), on reconstructions for *E. coli*[36] and *Bacillus subtilis*[36,37]. These formalisms succeeded at merging genome-scale metabolic networks together with comprehensive representations of macromolecular expression processes, enabling detailed exploration of the constraints that govern cellular growth on diverse environments. Despite the great advances for understanding cell physiology provided by these modeling formalisms, accuracy on phenotype predictions is compromised by the large number of parameters that are required (rate constants for transcriptional, translational, protein folding and degradation processes), with most of these not being readily available in the literature. Moreover, these models encompass processes that differ radically in their temporal scales (e.g., protein synthesis vs. metabolic rates) and their mathematical representation (presence of non-linear expressions in ME-models), requiring the implementation of more elaborate techniques for numerical simulation.

GECKO, a method for enhancement of GEMs with Enzymatic Constraints using Kinetic and Omics data, was developed in 2017 and applied to the consensus GEM for *S. cerevisiae*, Yeast7[38]. This method extends the classical FBA approach by incorporating a detailed description of the enzyme demands for the metabolic reactions in a network, accounting for all types of enzyme-reaction relations, including isoenzymes, promiscuous enzymes and enzymatic complexes. Moreover, GECKO enables direct integration of proteomics abundance data, if available, as constraints for individual protein demands, represented as enzyme usage pseudo-reactions, whilst all the unmeasured enzymes in the network are constrained by a pool of remaining protein mass. Additionally, this method incorporates a hierarchical and automated procedure for retrieval of kinetic parameters from the BRENDA database, which yielded a high coverage of kinetic constraints for the *S. cerevisiae* network. The resulting enzyme-constrained model, ecYeast7, was used for successful prediction of the Crabtree effect in wild-type and mutant strains of *S. cerevisiae* and cellular growth on diverse environments and genetic backgrounds, but also provided a simple framework for prediction of protein allocation profiles and study of proteomics data in a metabolic context. Furthermore, the model formed the basis for modeling yeast growth at different temperatures[39].

Since the first implementation of the GECKO method[38], its principles of enzyme constraints have been incorporated into GEMs for *B. subtilis*[40], *E. coli*[41], *B. coagulans*[42], *Streptomyces coelicolor*[43] and even for diverse human cancer cell-lines[2], showing the applicability of the method even for non-model organisms. Despite the rapid adoption of the method by the constraint-based modeling community, there is still a need for automating the model generation and enabling identification of kinetic parameters for less studied organisms.

In this work, we updated the GECKO toolbox to its 2.0 version, expanding its use it for building enzyme-constrained models (ecModels) for more organisms. Among other improvements, we generalized its structure to facilitate its applicability to a wide variety of GEMs, and we improved its parameterization procedure to ensure high coverage of kinetic constraints, even for poorly studied organisms. Additionally, we incorporated simulation utility functions, and developed an automated pipeline for updating ecModels, named ecModels container. This container is directly connected to the original sources of version-controlled GEMs and the GECKO toolbox, offering a continuously updated catalog of diverse ecModels.

## Results

**Community development of GECKO.** To ensure wide application and enable future development by the research community, we established the GECKO toolbox as open-source software, mostly encoded in MATLAB. It integrates modules for enhancement of GEMs with kinetic and proteomics constraints, automated retrieval of kinetic parameters from the BRENDA database (python module), as well as simulation utilities and export of ecModel files compatible with both the COBRA toolbox[44] and the COBRApy package[45]. The development of GECKO has been continuously tracked in a public repository (https://github.com/SysBioChalmers/GECKO) since 2017, providing a platform for open and collaborative development. The generation of output model files in.txt and SBML L3V1 FBC2[46] formats enabled the use of the ecYeastGEM[1] structure as a standard test to track the effects of any modifications in the toolbox algorithm through the use of the Git version control system, contributing to reproducibility of results and backwards compatibility of code.

Interaction with users of the GECKO toolbox and the ecYeastGEM model has also been facilitated through the use of the GECKO repository, allowing users to raise issues related with the programming of the toolbox or even about conceptual assumptions of the method, which has guided cumulative enhancements. Additionally, technical support for installation and utilization of the toolbox and ecYeastGEM is now provided through an open community chat room (available at: https://gitter.im/SysBioChalmers/GECKO), reinforcing transparent and continuous communication between users and developers.

**New additions to the GECKO toolbox**. The previous implementation of the GECKO method in GECKO 1.0 significantly improved phenotype predictions for *S. cerevisiae*'s metabolism under a wide variety of genetic and environmental perturbations[38]. However, its development underscored some issues, in particular that quantitative prediction of the critical dilution rate and exchange fluxes at fermentative conditions are highly sensitive to the distribution of incorporated kinetic parameters. Although *S. cerevisiae* is one of the most studied eukaryote organisms, not all reactions included in its model have been kinetically characterized. Therefore, a large number of $k_{cat}$ numbers measured for other organisms (48.35%), or even non-specific to their reaction mechanism (56.03% of $k_{cat}$ values found by introduction of wildcards into E.C. numbers) were needed to be incorporated, in order to fill the gaps in the available data for the reconstruction of the first *S. cerevisiae* ecModel, ecYeast7. Moreover, detailed manual curation of $k_{cat}$ numbers was needed for several key enzymes in order to achieve biologically meaningful predictions.

As the BRENDA database[47] is the main source of kinetic parameters for GECKO, all of the available $k_{cat}$ and specific activity entries for non-mutant enzymes were retrieved. In total, 38,280 entries for 4130 unique E.C. numbers were obtained and classified according to biochemical mechanisms, phylogeny of host organisms and metabolic context (Brenda kinetic data analysis section in the Supplementary Information File 1), in order to assess significant differences in distributions of kinetic parameters. This analysis showed that not all organisms have been equally studied. While entries for *H. sapiens*, *E. coli*, *R. norvegicus*, and *S. cerevisiae* account for 24.02% of the total, very few kinetic parameters are available for most of the thousands of organisms present in the database, showing a median of 2 entries per organism (Fig. 1a). The analysis also showed that kinetic activity can differ drastically, spanning several orders of magnitude even for families of enzymes with closely related biochemical mechanisms (Fig. 1b). Finally, it was also observed that $k_{cat}$ distributions for enzymes in the central carbon and energy metabolism differ significantly from those in other metabolic contexts across phylogenetic groups of host organisms (life kingdoms, according to the KEGG phylogenetic tree[48]), even without filtering the dataset for entries reported exclusively for natural substrates, as previously done by other studies[49] (Fig. 1c).

In the new version of the GECKO toolbox (GECKO 2.0), a modified set of hierarchical $k_{cat}$ matching criteria was implemented to address how $k_{cat}$ numbers depend on biochemical mechanisms, metabolic context and phylogeny of host organisms. The modified parameterization procedure enables the incorporation of kinetic parameters that have been reported as *specific activities* in BRENDA when no $k_{cat}$ is found for a given query (as the specific activity of an enzyme is defined as its $k_{cat}$ over its molecular weight), adding 8,118 new entries to the catalog of kinetic parameters in the toolbox. A phylogenetic distance-based criterion, based on the phylogenetic tree available in the KEGG database[48], was introduced for cases in which no organism-specific entries are available for a given query in the kinetic parameters dataset. Specifically, where GECKO 1.0 chooses $k_{cat}$ available in BRENDA regardless of organism, GECKO 2.0 chooses the values available in BRENDA for the phylogenetically closest organism by iteratively introducing a wildcard into the E.C. number, as exemplified in the Brenda kinetic data analysis section in the Supplementary Information File 1 "EC3.x.x.x", and estimating the phylogenetic distance. The new $k_{cat}$ matching algorithm, including the estimation of the phylogenetic distance, and its comparison with the predecessor are shown in the supplementary methods section in Supplementary File 1.

In order to assess the impact of the modified $k_{cat}$ assignment algorithm on an ecModel, ecYeast7 was reconstructed using both the first and GECKO 2.0. A classification of the matched $k_{cat}$ values according to the new matching algorithm is provided in Fig. 1d, showing the amount of values chosen from the phylogenetically closest organisms. The incorporation of specific activity values in the parameter catalog increased the number of kinetic parameters matched to complete E.C. numbers (no added wildcards) from 1432 to 2696 (Fig. 1e). Moreover, the implementation of the phylogenetic distance-based criterion yielded a distribution of kinetic parameters that showed no significant differences when compared to the values reported in BRENDA for all fungi species, in contrast to the kinetic profile matched by the previous algorithm (*P*-values $2.1 \times 10^{-11}$ and $3.9 \times 10^{-8}$, when compared to the BRENDA fungi and *S. cerevisiae* distributions, respectively, under a two-tailed Kolmogorov–Smirnov test) (Fig. 1f). The quality of phenotype predictions for the ecYeast7 model enhanced by GECKO 2.0 was evaluated by simulation of batch growth in 19 different environments, with an average relative error of 23.97% when compared to experimental data (Fig. 1g); in contrast, its GECKO 1.0 counterpart yielded an average relative error of 32.07%.

The introduction of manually curated $k_{cat}$ numbers in a metabolic network has been proven to increase the quality of phenotype predictions for *S. cerevisiae*[22,25,38]; nevertheless, this is an intensive and time-consuming procedure that is hard to ensure for a large number of models subject to continuous modifications. In order to ensure applicability of the GECKO method to any standard GEM, a unified procedure for curation of kinetic parameters was developed based on parameter sensitivity analysis. For automatically generated ecModels that are not able to reach the provided experimental value for maximum batch growth rate, an automatic module performs a series of steps in which the top enzymatic limitation on growth rate is identified through the quantification of enzyme control coefficients. For such enzymes, the E.C. number is obtained and then its correspondent $k_{cat}$ value is substituted by the highest one available in BRENDA for the given enzyme class. This procedure iterates until the specific growth rate predicted by the model reaches the provided experimental value.

Finally, as the first version of the toolbox relied on the structure and nomenclature of the model Yeast7, its applicability to other reconstructions was not possible in a straightforward way. In order to provide compatibility with any other GEM, based on COBRA[44] or RAVEN[50] formats, all of the organism-specific parameters required by the method (experimental growth rate, total protein content, organism name, names and identifiers for some key reactions, etc.) can be provided in a single MATLAB initialization script, minimizing the modifications needed for the generation of a new ecModel.

**ecModels container is an automatically updated repository**. Several GEMs that have been published are still subject to continuous development and maintenance[1–3,5,6], this renders GEMs

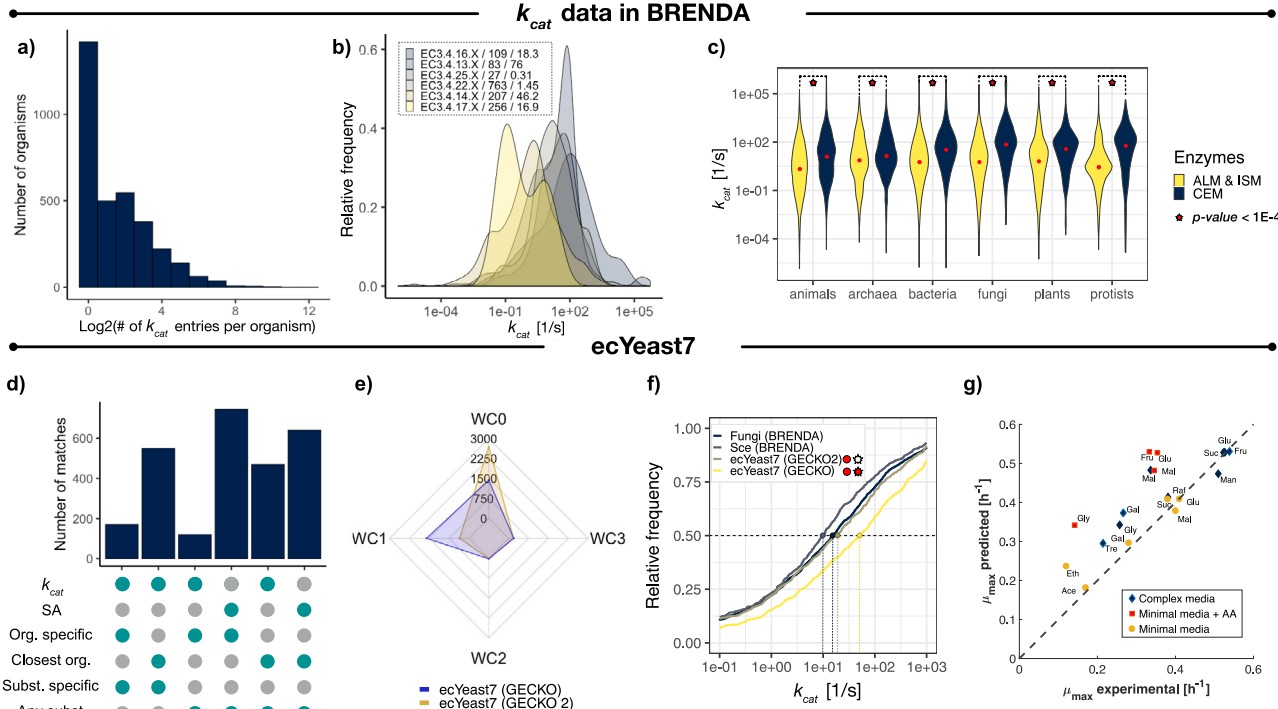

**Fig. 1 $k_{cat}$ distributions in BRENDA and ecYeast7. a** Number of $k_{cat}$ entries in BRENDA per organism. **b** $k_{cat}$ distributions for closely related enzyme families. Sample size and median values (in s$^{-1}$) are shown after each family identifier. **c** $k_{cat}$ distributions for enzymes in BRENDA by metabolic context and life kingdoms. Median values are indicated by red dots in each distribution, statistical significance (under a one-sided Kolmogorov–Smirnov test) is indicated by red stars for each pair of distributions for a given kingdom. CEM—central carbon and energy metabolism; ALM—Amino acid and lipid metabolism; ISM—intermediate and secondary metabolism. Computed P-values are $2.8 \times 10^{-27}$ for animals; $3.85 \times 10^{-5}$ for archaea; $1.62 \times 10^{-92}$ for bacteria; $1.024 \times 10^{-30}$ for fungi; $2.36 \times 10^{-16}$ for plants and $4.75 \times 10^{-21}$ for protists. **d** Number of $k_{cat}$ matches in ecYeast7 per assignment category (GECKO 2.0). **e** Comparison of the number of $k_{cat}$ matches for E.C. numbers with 0, 1, 2, and 3 introduced wildcards by GECKO 2.0 and GECKO $k_{cat}$ matching algorithms. **f** Cumulative $k_{cat}$ distributions for: all *S. cerevisiae* entries in BRENDA, all entries for fungi in BRENDA, ecYeast7 enhanced by GECKO and ecYeast7 enhanced by GECKO 2.0. Colored points and vertical dashed lines indicate the median value for each distribution. Statistical significance under a two-sided Kolmogorov–Smirnov test of the matched $k_{cat}$ distributions when compared to all entries for *S. cerevisiae* and fungi, is shown with red circles and stars, respectively. P-values below $1 \times 10^{-2}$ are indicated with red. Computed P-values are 0.538 for the comparison between GECKO2 vs. all fungi, $2.7 \times 10^{-3}$ for GECKO2 vs. *S. cerevisiae*, $3.9 \times 10^{-8}$ for GECKO vs. all fungi and, $2.1 \times 10^{-11}$ for GECKO vs. the *S. cerevisiae* entries. **g** Prediction of batch maximum growth rates on diverse media with ecYeast7 enhanced by GECKO 2.0. Glu—glucose, Fru—fructose, Suc—sucrose, Raf— raffinose, Mal—maltose, Gal—galactose, Tre—trehalose, Gly—glycerol, Ace—acetate, Eth —ethanol. Source data are provided in Source Data: Data Source file 1.

to be dynamic structures that can change rapidly. In order to integrate such continuous updates into the enzyme-constrained version of a model in an organized way, an automated pipeline named *ecModels container* was developed.

The ecModels container is a continuous integration implementation whose main functionality is to provide a catalog of ecModels for several relevant organisms that are automatically updated every time a modification is detected either in the original GEM source repository or in the GECKO toolbox, i.e., new releases in their respective repositories. The pipeline generates ecModels in different formats, including the standard SBML and MATLAB files, and stores them in a container repository (https://github.com/SysBioChalmers/ecModels) in a version-controlled way, requiring minimal human interaction and maintenance. The GECKO toolbox ensures the creation of functional and calibrated ecModels that are compatible with the provided experimental data (maximum batch growth rate, total protein content of cells, and exchange fluxes at different dilution rates as an optional input). This whole computational pipeline is illustrated in Fig. 2. Further description of the ecModels container pipeline functioning is included in the "Methods" section.

**A catalog of new ecModels.** Following the aforementioned additions to the GECKO toolbox, that have allowed its

generalization, we used the toolbox for the reconstruction of four new ecModels from previously existing high-quality metabolic network reconstructions: *i*Yali4, for the oleaginous yeast *Yarrowia lipolytica*[5]; *i*SM996, for the thermotolerant yeast *Kluyveromyces marxianus*[6]; *i*ML1515, for the widely studied bacterium *E. coli*[4]; and Human1, being the latest and largest network reconstruction available for studying *H. sapiens* metabolism[2]. For the microbial models, all model parameters were calibrated according to the provided experimental data, generated by independent studies[4,51–53], yielding functional ecModels ready for simulations. Size metrics for these models can be seen in Table 1.

These ecModels, together with ecYeastGEM, are hosted in the ecModels container repository for their continuous and automated update every time that a version change is detected either in the original model source or in the GECKO repository. In the case of microbial species, two different model structures are provided: *ecModel*, which has unbounded individual enzyme usage reactions ready for incorporation of proteomics data; and *ecModel_batch* in which all enzyme usage reactions are connected to a shared protein pool. This pool is then constrained by experimental values of total protein content, and calibrated for batch simulations using experimental measurements of maximum batch growth rates on minimal glucose media, thus providing a functional ecModel structure ready for simulations.

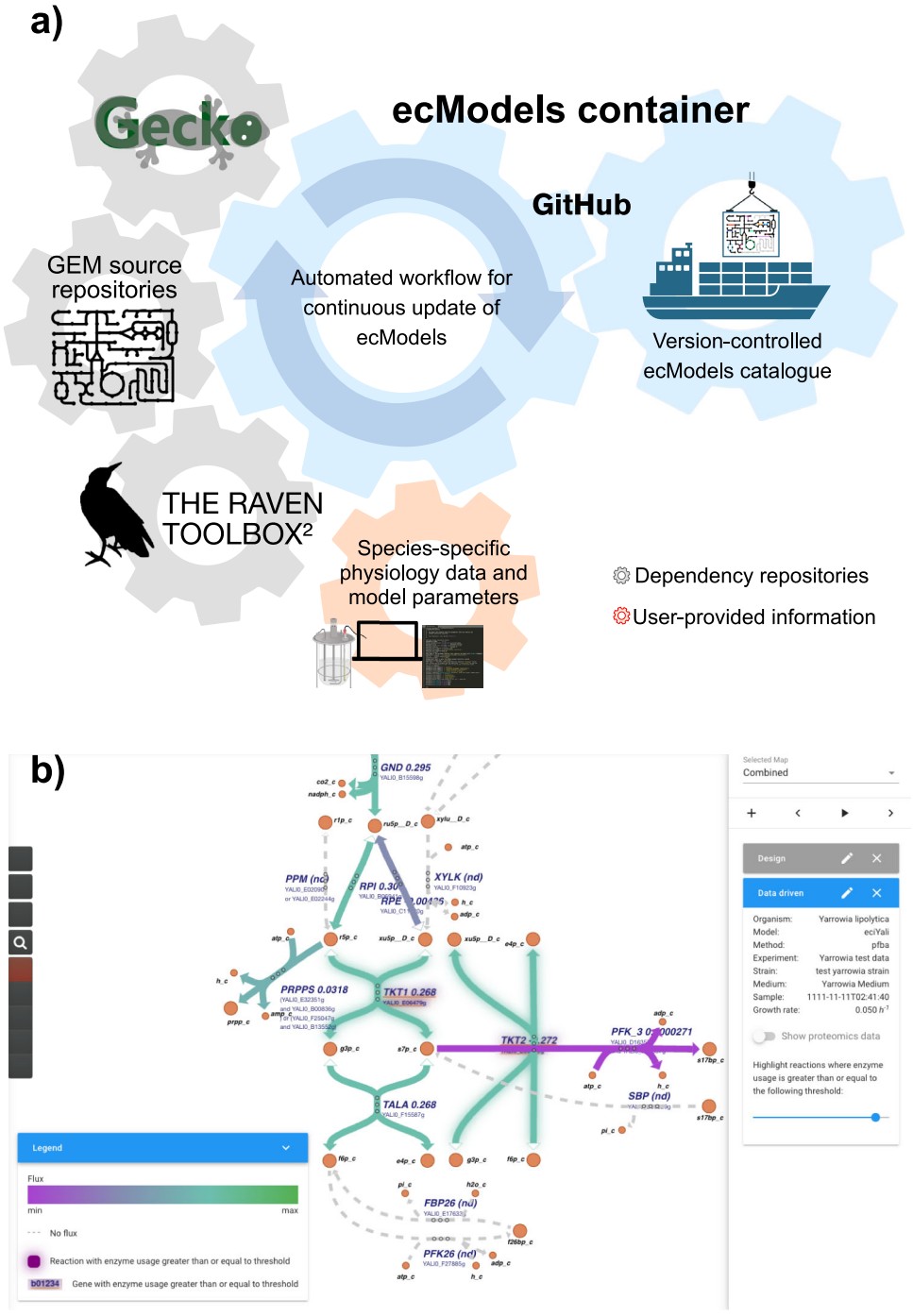

**Fig. 2 Extending utilization of ecModels. a** ecModels container: Integrated pipeline for continuous and automated update of ecModels. **b** Implementation of GECKO simulations in the Caffeine platform (https://caffeine.dd-decaf.eu/) for visualization of enzyme usage. The color of the arrows corresponds to the value of the corresponding fluxes. Genes or reactions connected to enzymes with a usage above 90% are highlighted with a glow around the corresponding text or arrow, respectively. The chosen usage threshold to highlight can be tuned with the slider on the right.

For ecHumanGEM just the unbounded ecModel files are provided, as this is a general network of human metabolism, containing all reactions from any kind of human tissue or cell type for which evidence is available, and therefore not suitable for numerical simulation. As *H. sapiens* is the most represented organism in the BRENDA database, accounting for 11% of the total number of available $k_{cat}$ values (Brenda kinetic data analysis section in the Supplementary Information File 1), kinetic parameters from other organisms were not taken into account for its enhancement with enzyme constraints. ecHuman1

provides the research community with an extensive knowledge base that represents a complete and direct link between genes, proteins, kinetic parameters, reactions and metabolites for human cells in a single model structure, subject to automated continuous update by the *ecModels container* pipeline.

**Visualization of GECKO simulations in the Caffeine platform**. We implemented simulations with ecModels in Caffeine, an open-source software platform for cell factory design. Caffeine, publicly available at http://caffeine.dd-decaf.eu, allows user-

**Table 1 Size metrics summary for the ecModels catalog.**

| Original GEMs | | | | | |
|---|---|---|---|---|---|
| Organism | *S. cerevisiae* | *Y. lipolytica* | *K. marxianus* | *E. coli* | *H. sapiens* |
| Model ID | yeastGEM_8.3.3 | *i*Yali4 | *i*SM996 | *i*ML1515 | Human1 |
| Reactions | 3963 | 1924 | 1913 | 2711 | 13101 |
| Metabolites | 2691 | 1671 | 1531 | 1877 | 8400 |
| Genes | 1139 | 847 | 996 | 1516 | 3628 |
| Enzyme-constrained GEMs | | | | | |
| Model ID | ecYeastGEM | ec*i*Yali | ec*i*SM996 | ec*i*ML1515 | ecHumanGEM |
| Reactions | 8028 | 3881 | 5334 | 6084 | 46259 |
| Metabolites | 4153 | 1880 | 2064 | 2334 | 12191 |
| Enzymes | 965 | 647 | 716 | 1259 | 3224 |
| Enzyme coverage | 84.72% | 76.39% | 71.89% | 83.05% | 88.86% |
| Reactions w/$k_{cat}$ | 3771 | 1586 | 2891 | 2562 | 27014 |
| Reactions w/ Isoenzymes | 504 | 205 | 532 | 456 | 3791 |
| Promiscuous Enzymes | 572 | 324 | 469 | 673 | 2184 |
| Enzyme complexes | 252 | 75 | 27 | 383 | 756 |

friendly simulation and visualization of flux predictions made by genome-scale metabolic models. Several standard modeling methods are already included in the platform, such as $^{13}$C fluxomics data integration, and simulation of gene deletion and/ or overexpression, to interactively explore strain engineering strategies. In order to allow for GECKO simulations, we added a new feature to the platform for uploading enzyme-constrained models and absolute proteomics data. Additionally, we added a simulation algorithm that recognizes said models, and overlays the selected proteomics data on them, leaving out data that makes the model unable to grow at a pre-specified growth rate. After these inclusions to the platform, enzyme usage can now be computed on the fly and visualized on metabolic maps (Fig. 2b), to identify potential metabolic bottlenecks in a given condition. The original proteomics data can be visualized as well, to identify if the specific bottleneck is due to a lack of enzyme availability, or instead due to an inefficient kinetic property. This will suggest different metabolic engineering strategies to the user: if the problem lies in the intracellular enzyme levels, the user can interpret this as a recommendation for overexpressing the corresponding gene, whereas if the problem lies in the enzyme efficiency, the user could assess introducing a heterologous enzyme as an alternative.

**GECKO simulation utilities**. As ecModels are defined in an irreversible format and incorporate additional elements such as enzymes (as new pseudo-metabolites) and their usages (represented as pseudo-reactions), they might sometimes not be directly compatible with all of the functionalities offered by currently available constraint-based simulation software[44,45,50,54,55]. We therefore added several new features to the GECKO toolbox that allow the exploration and exploitation of ecModels. These include utilities for: (1) basic simulation and analysis purposes, (2) accessible retrieval of kinetic parameters, (3) automated generation of condition-dependent ecModels with proteomic abundance constraints, (4) comparative flux variability analysis between a GEM and its ecModel counterpart, and (5) prediction of metabolic engineering targets for enhanced production with an implementation of the FSEOF method[56] for ecModels. Detailed information about the inputs and outputs for each utility can be found on their respective documentation, available at: https://github.com/SysBioChalmers/GECKO/tree/master/geckomat/utilities. All of these utilities were developed in MATLAB due to their dependency on some RAVEN toolbox functions[50].

**Predicting microbial proteome allocation in multiple environments**. In order to test the quality of the phenotype predictions of an ecModel automatically generated by the *ecModels container* pipeline, batch growth under 11 different carbon sources was simulated with ec*i*ML1515 for *E. coli*. Figure 3a shows that, for all carbon sources, growth rates were predicted at the same order of magnitude as their corresponding experimental measurements, with the most accurate predictions obtained for growth on D-glucose, mannose and D-glucosamine. Furthermore, batch growth rate and protein allocation predictions, using no exchange flux constraints, were compared between ec*i*ML1515 and the *i*JL1678 ME-model[32], the latter accounting for both metabolism and macromolecular expression processes. The sum squared error (SSE) for batch growth rate predictions across the 11 carbon sources using ec*i*ML1515 was 0.27, a drastic improvement when compared to the 1.21 SSE of *i*JL1678 ME-model predictions[32]. Figure 3b shows the predicted total proteome needed by cells to sustain the provided experimental growth rates for the same 11 environments. Notably ec*i*ML1515 predicts values that lie within the range of predictions of the *i*JL1678 ME-model (from the optimal to the generalist case) for 10 out of the 11 carbon sources (see "Methods" for simulation details). This shows that the new version of the GECKO toolbox ensures the generation of functional ecModels that can be readily used for simulation of metabolism, due to its systematic parameter flexibilization step, which reduces the need of extensive manual curation for new ecModels. Furthermore, *i*ML1515 is a model available as a static file at the BiGG models repository[57]; therefore, its integration to the ecModels container for continuous update demonstrates the flexibility of our pipeline, regarding compatibility with original GEM sources, which can be provided as a link to their *git*-based repositories or even as static URLs.

**Proteomics constraints refine phenotype predictions for multiple organisms and conditions**. The previously mentioned module for integration of proteomics data generates a condition-dependent ecModel with proteomics constraints for each condition/replicate in a provided dataset of absolute protein abundances [mmol/gDw]. Even though absolute quantification of proteins is becoming more accessible and integrated into systems biology studies[58–62], a major caveat of using proteomics data as constraints for quantitative models is their intrinsic high biological and technical variability[63], therefore some of the incorporated data constraints need to be loosened in order to obtain functional ecModels. When needed, additional condition-

**Fig. 3 Comparison of predictive capabilities between ec*i*ML1515 and ME-*i*JL1678 for *E. coli*. a** Maximum batch growth rate predictions on minimal media with diverse carbon sources, with an average relative error for ec*i*ML1515 of 34,43%, and an $R^2$ of 0.196. The sum of squared errors when compared to experimental values are 0.2785 for ec*i*ML1515 and 1.21 for ME-*i*JL1678. **b** Prediction of total protein content in the cell by ec*i*ML1515 and ME-*i*JL1678 using the optimal and generalist approaches. Source data are provided in Source Data: Data Source file 1.

dependent exchange fluxes of byproducts can also be used as constraints in order to limit the feasible solution space. A detailed description of the proteomics integration algorithm implemented in GECKO is given in the supplementary methods section in the Supplementary Information File 1.

The new proteomics integration module was tested on the three ecModels for budding yeasts available in ecModels container (ecYeastGEM, ec*i*Yali, ec*i*SM996). We measured absolute protein abundances for *S. cerevisiae, Y. lipolytica* and *K. marxianus*, grown in chemostats at 0.1 h$^{-1}$ dilution rate and subject to several experimental conditions (high temperature, low pH and osmotic stress with KCl)[64], and incorporated these data into the ecModels as upper bounds for individual enzyme usage pseudo-reactions. Then, exchange fluxes for $CO_2$ and oxygen corresponding to the same chemostat experiments were used as a comparison basis to evaluate quality of phenotype predictions. For each organism- condition pair, 3 models were generated and compared in terms of predictions: a pure stoichiometric metabolic model, an enzyme-constrained model with a limited shared protein pool, and an enzyme-constrained model with proteomics constraints. It was found that the addition of the enzyme pool constraint enables major reduction of the relative error in prediction of gaseous exchange fluxes in some of the studied conditions. Additionally, the incorporation of individual protein abundance constraints improves even further the predictive accuracy of gaseous exchanges, for 5 out of the 11 evaluated cases (Fig. 4a–c). Although only a trend and not a significant improvement, it would be of interest, in the future, to run further analyses that include more proteomics datasets.

The impact of incorporating enzyme and proteomics constraints on intracellular flux predictions was further assessed by mapping all condition-dependent flux distributions from the tested ecModels to their corresponding reactions in the original GEMs. In general, metabolic flux distributions showed high similarity when comparing ecModel to GEM predictions (Supplementary Fig. 1), as 70–90% of the active reaction fluxes were predicted within the interval of $0.5 < \text{fold-change} < 2 \left( \text{FC} = \frac{v_i^{\text{ecModel}}}{v_i^{\text{GEM}}} \right)$ across all conditions (Supplementary Fig. 2A–C, Source Data: Data Source File 2). In addition, principal component analysis on

absolute enzyme usage profiles predicted by ecModels revealed that, at low dilution rates, predictions of enzyme demands are mostly defined by the selected set of imposed constraints (shared protein pool vs. proteomics constraints) rather than by environmental condition, i.e., exchange fluxes (Supplementary Fig. 2D–F). However, more straightfroward comparison of the models' predictions, by pairwise comparison of predicted absolute enzyme usage profiles, showed that 60–80% of the predicted enzyme usages lie within a range of $0.5 < \text{fold-change} < 2$, when comparing ecModels predictions with and without proteomics constraints, across organisms and conditions (Fig. 4d, Supplementary Fig. 2G–I, and Data Source File 2). It was observed that the incorporation of proteomics constraints induces a drastic differential use for a considerable amount of enzymes, as 12–21% of enzyme usages were predicted as either enabled or disabled by these constraints across all the simulated conditions, showing slight enrichment for enabled alternative isoenzymes for already active reactions (Data Source File 2). This suggests that upper bounds on enzyme usages induce differentiated utilization of isoenzymes, reflecting well why isoenzymes have been maintained throughout evolution.

The explicit inclusion of enzymes into GEMs by the GECKO method enables prediction of enzyme demands at the protein, reaction and pathway levels. Total protein burden values predicted by ecModels for several relevant metabolic superpathways (central carbon and energy metabolism, amino acid metabolism, lipid and fatty acid metabolism, cofactor and vitamin metabolism and nucleotide metabolism, according to the KEGG metabolic subsystems[48]), showed that central carbon and energy metabolism is the most affected sector in the ecYeastGEM network by integration of proteomics constraints, as protein burden predictions were higher, at least by 20%, for 3 out of the 4 simulated conditions when compared with predictions of the ecYeastGEM without proteomics data (Fig. 4e).

Relative enzyme usages, estimated as predicted absolute enzyme usage over enzyme abundance for all of the measured enzymes in an ecModel $\left( \frac{e_i}{[E_i]} \right)$, can be understood as the saturation level of enzymes in a given condition. In order to analyze the metabolic mechanisms underlying long-term adaptation to stress in budding yeasts, relative enzyme usage profiles

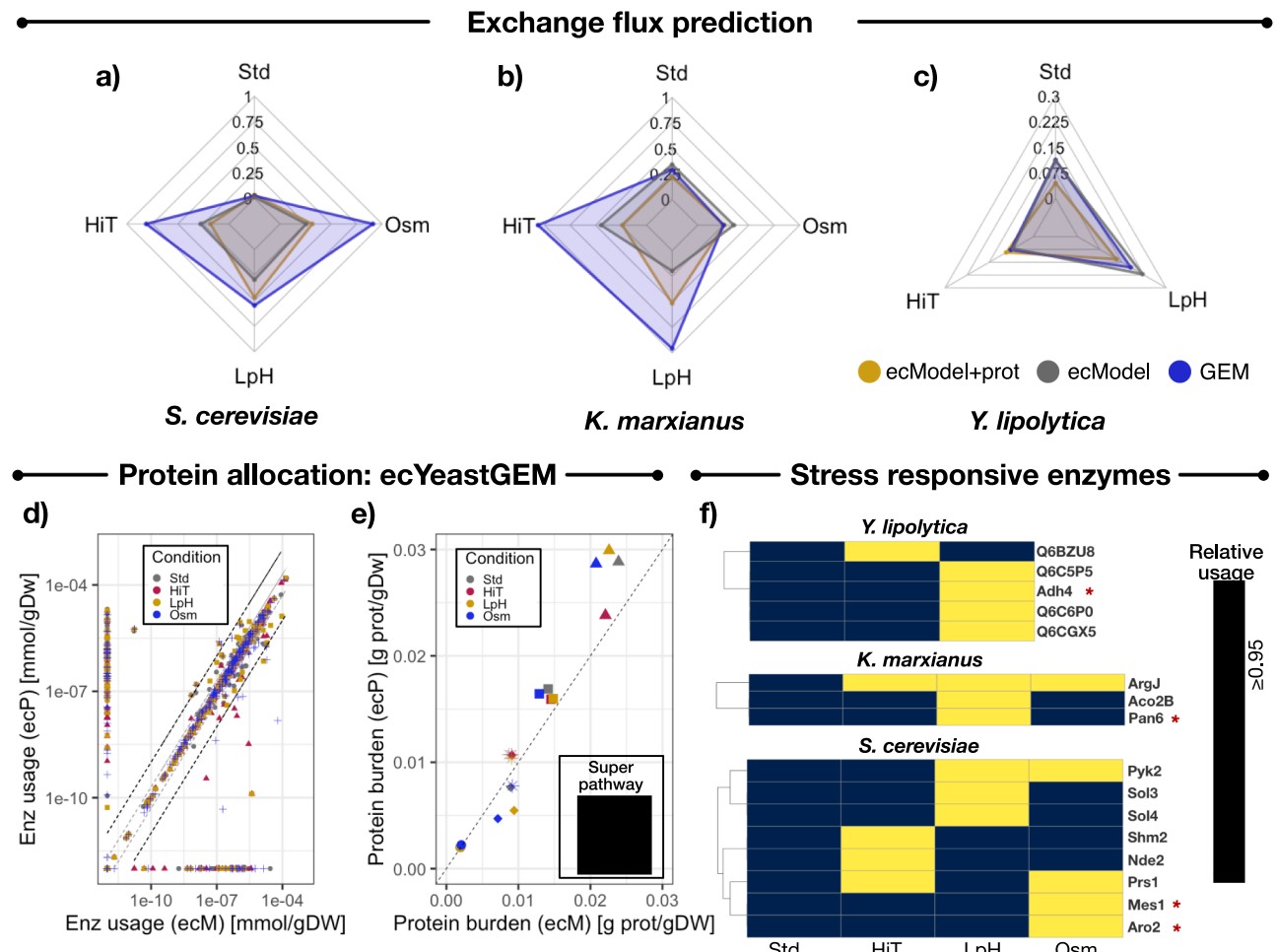

**Fig. 4 Evaluation of proteomics-constrained ecModels.** Comparison of median relative error in prediction of exchange fluxes for $O_2$ and $CO_2$ by GEMs, ecModels and proteomics-constrained ecModels across diverse conditions (chemostat cultures at 0.1 h$^{-1}$ dilution rate) for **a** *S. cerevisiae*, **b** *K. marxianus*, **c** *Y. lipolytica*. **d** Comparison of absolute enzyme usage profiles [mmol/gDw] predicted by ecYeastGEM (ecM) and ecYeastGEM with proteomics constraints (ecP) for several experimental conditions. The region between the two dashed gray lines indicates enzyme usages predicted in the interval $0.5 \leq E_i^{ecP}/E_i^{ecM} \leq 2$, the region between the two dashed black lines indicates enzyme usages predicted in the interval $0.1 \leq E_i^{ecP}/E_i^{ecM} \leq 10$, when comparing the two ecModels. **e** Protein burden for different superpathways predicted by ecYeastGEM (ecM) and ecYeastGEM with proteomics constraints (ecP). **f** Highly saturated enzymes at different stress conditions for *S. cerevisiae*, *K. marxianus*, and *Y. lipolytica* predicted by their corresponding ecModels constrained with proteomics data. Yellow cells indicate condition-responsive enzymes (relative usage ≥ 0.95). Red asterisks indicate enzymes conserved as single copy orthologs across the three yeast species. Std—Reference condition, HiT—high-temperature condition, LpH—Low pH condition, Osm—Osmotic stress condition, AA—amino acid metabolism, NUC—nucleotide metabolism, CEM—central carbon and energy metabolism, CofVit—cofactor and vitamin metabolism, Lip—lipid and fatty acid metabolism. Source data are provided in Source Data: Data Source File 2.

were computed from all the previous simulations of ecModels with proteomics constraints. Enzymes that display fold-changes higher than 1 for both absolute abundance and their saturation level, when comparing predicted usage profiles between stress and reference conditions, suggest regulatory mechanisms on individual proteins that contribute to cell growth on the anlyzed stress condition. Figure 4f shows all of the enzymes that were identified as responsive to environmental stress in this study, displaying enrichment for enzymes involved in biosynthesis of diverse amino acids and folate metabolism.

A further mapping of all enzymes in these ecModels to a list of 2,959 single copy protein-coding gene orthologs across the three yeast species[64] found 310 core proteins across these ecModels. Principal component analysis revealed that variance on absolute enzyme usages and abundance profiles for these core proteins is mostly explained by differences in the metabolic networks of the different species rather than by environmental conditions (Supplementary Fig. 3B, C), reinforcing previous results

suggesting that, despite being phylogenetically related, their long-term stress responses at the molecular level have evolved independently after their divergence in evolutionary history[64].

**Exploring the solution space reduction.** A major limitation in the use of GEMs is the high variability of flux distributions for a given cellular objective when implementing flux balance analysis, as this requires solving largely underdetermined linear systems through optimization algorithms[15,65]. This limitation has usually been overcome with incorporation of measured exchange fluxes as constraints. However, these data are typically sparse in the literature. Previous studies explored the drastic reduction in flux variability ranges of ecModels for *S. cerevisiae* and 11 human cell-lines when compared to their original GEMs due to the addition of enzyme constraints[1,2,38]. However, the irreversible format of ecModels (forward and backwards reactions are split in order to account for enzyme demands of both directions) hinders their compatibility with the flux variability analysis (FVA) functions

already available in COBRA[44] and RAVEN[50] toolboxes. As a solution to this, an FVA module was integrated to the utilities repertoire in GECKO, whose applicability has been previously tested on studies with ecModels for *S. cerevisiae*[1] and human cell lines[2]. This module contains the necessary functions to perform FVA on any set of reactions of an ecModel, enabling also a direct comparison of flux variability ranges between an ecModel and its GEM counterpart in a consistent way (supplementary methods section in the Supplementary Information File 1).

The FVA utility was applied on three different ecModels of microbial metabolism and their correspondent GEMs (*i*ML1515, *i*Yali4, and *i*SM996). In all cases the FVA comparisons were carried out for both chemostat and batch growth conditions in order to span different degrees of constraining of the metabolic networks ($0.1\,h^{-1}$ dilution rate and minimal glucose uptake rate fixed for chemostat conditions; biomass production fixed to experimental measurements of $\mu_{max}$ and unconstrained uptake of minimal media components, for batch conditions). Cumulative distributions for flux variability ranges for all explored ecModels and GEMs are shown in Fig. 5, in which it can be seen that median flux variability ranges are much reduced for all ecModels and conditions, especially at high growth rates where enzyme constraints reduce the variability range 5–6 orders of magnitude when compared to pure GEMs. The cumulative distributions also show a major reduction in the amount of totally variable fluxes (reactions that can carry any flux between −1000 to 1000 mmol/gDwh), which are an indicator of undesirable futile cycles present in the network due to lack of thermodynamic and enzyme cost information[66–68]. For high growth rates, the amount of totally variable fluxes accounts for 3–12% of the active reactions in the analyzed GEMs, in contrast to their corresponding ecModels in which such extreme variability ranges are completely absent.

Further analysis of the FVA results revealed that a reduction of at least 95% of the variability range was achieved for more than 90% of all fluxes of active reactions at high growth rates in all

ecModel. Interestingly, the aforementioned flux variability metrics were overall improved even for the chemostat conditions, despite a higher degree of constraining (fixed low growth rate and optimal uptake rate), which restrains these models to an energy efficient respiratory mode (Data Source File 3).

## Discussion

Here, we demonstrated how enzyme-constrained models for diverse species significantly improve simulation performance compared to traditional GEMs. Furthermore, to enable the community to easily adapt this modeling approach, we upgraded the GECKO toolbox for enhancement of genome-scale models with enzyme and omics constraints to its version 2.0. Major improvements on the $k_{cat}$ matching algorithm were incorporated into the toolbox, based on phylogenetic distance between the modeled organism and the host organisms for data queries, and an automated curation of $k_{cat}$ numbers for over-constrained models were incorporated into the toolbox. Major refactoring of the GECKO toolbox enabled a generalization of the method, allowing the creation of high-quality ecModels for any provided functional GEM with minimal need for case-specific introduction of new code. Additionally, several utility functions were integrated into the toolbox in order to enable basic simulation purposes, accessible retrieval of enzyme parameters, integration of proteomics data as constraints, flux variability analysis and prediction of gene targets for enhanced production of metabolites. Overall, it was shown that these enhancements to the GECKO toolbox improve the incorporation of kinetic parameters into a metabolic model, yielding ecModels with biologically meaningful kinetic profiles without compromising accuracy on phenotype predictions.

Two major limitations of the first version of the GECKO toolbox were its specific customization to the *S. cerevisiae* model, Yeast7, and the need of extensive manual curation for generating an ecModel suited for FBA simulations; thus, its applicability to

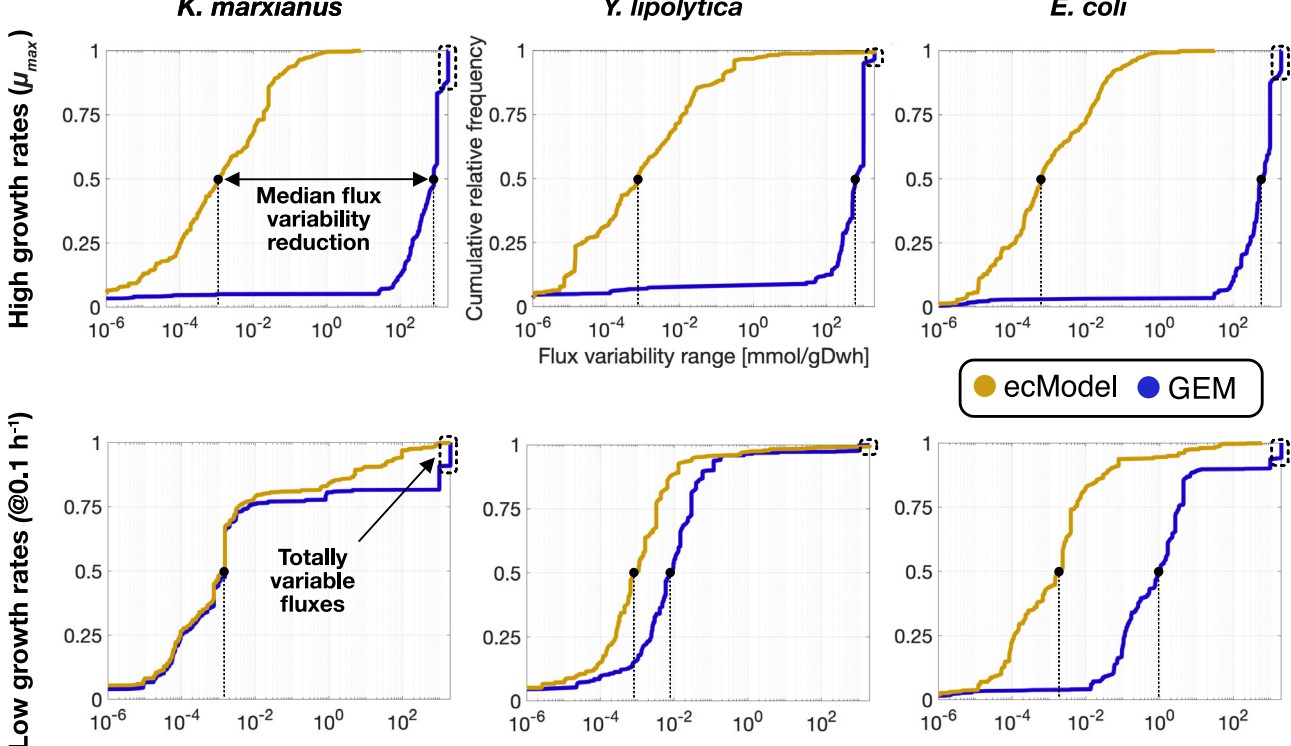

**Fig. 5 Cumulative distributions of flux variability ranges for *i*SM996, *i*Yali4 and *i*ML1515 compared to their respective enzyme-constrained versions at low and high growth rates.** Source data are provided in the Source Data: Data Source File 3.

other GEMs was not a straightforward procedure. To overcome these limitations, we generalized the code with the aim of making GECKO a model-agnostic tool. The development of a procedure for automatic curation of kinetic parameters enabled the generation of functional ecModels with minimal requirements for experimental data. Recently, ecModels for 11 human cancer cell-lines were generated with this automated procedure, using Human1 as a model input and RNAseq datasets together with the tINIT algorithm[10] to generate cell-line specific networks[2]. These ecModels were used for the prediction of cellular growth and metabolite exchange rates at different levels of added constraints, resulting in remarkable improvements in accuracy when compared with predictions of their original GEMs. This highlights one of the main advantages of ecModels: their capability of yielding biologically meaningful phenotype predictions without an excessive dependency on exchange fluxes as constraints.

In order to further showcase the functionality of the GECKO toolbox 2.0, a family of new high-quality ecModels were generated for *E. coli, Y. lipolytica, K. marxianus* and *H. sapiens*, based on the original GEMs *i*ML1515, *i*Yali4, *i*SM996 and Human1, respectively. Furthermore, we generated a self-hosted pipeline for continuous and automated generation and update of ecModels, *ecModels container*, so that each of the currently available ecModels (ecYeastGEM, ec*i*ML1515, ec*i*Yali, ec*i*SM996, and ecHuman1) are integrated to it, providing a version-controlled and continuously updated repository for high-quality ecModels. Moreover, the implemented automation facilitates the application of the GECKO method to other organisms for which sufficient data is available.

Absolute proteomics measurements for the budding yeasts *S. cerevisiae, K. marxianus* and *Y. lipolytica* grown under multiple environmental conditions, were incorporated as constraints into their ecModels by using the proteomics integration module added to the GECKO toolbox. Analysis of metabolic flux distributions revealed that net reaction fluxes predicted by GEMs are not significantly affected by the incorporation of kinetic and proteomics constraints, however, the explicit integration of enzymes into ecModels extends the range of predictions of classical FBA and enables computation of enzyme demands at the reaction and pathway levels. It was found that incorporation of proteomics constraints does not affect enzyme demand predictions significantly for most of the active enzymes at low dilution rates across the simulated conditions. However, we observed that a diversified utilization of isoenzymes, enforced by proteomics constraints, increases the predicted total protein mass allocated to central carbon and energy metabolism, in comparison to optimal enzyme allocation profiles. This result suggests the relevance of metabolic robustness in contrast to optimal protein utilization for microbial growth under environmental stress and nutrient-limited conditions.

Incorporation of proteomics data allows the use of ecModels as scaffolds for systems-level studies of metabolism, providing a tool for uncovering metabolic readjustments induced by genetic and environmental perturbations, which might be difficult to elucidate by purely data-driven approaches, specially at conditions of relatively low changes at the transcript[69] and protein levels[64]. For all studied stress conditions in this study, we identified upregulated proteins (increased abundance) that are needed to operate at high saturation levels in stress conditions, while showing low usage at reference conditions, creating lists of potential gene amplification targets for enhancing stress tolerance in three industrially relevant yeast species (Source Data: Data Source File 2). Upregulation and high saturation of enzymes in amino acid and folate metabolism were found to be common across the studied organisms and stress conditions (Supplementary Fig. 3D and Source Data: Data Source File 2). These results suggest that

yeast cells display enzyme expression profiles that provide them with metabolic robustness for microbial growth under stress and nutrient-limited conditions, in contrast to an optimal protein allocation strategy that prioritizes expression of the most efficient and non-redundant enzymes.

Our results on drastic reduction of median flux variability ranges and the number of totally unbounded fluxes for ec*i*Yali, ec*i*SM996, and ec*i*ML1515, together with previous studies[1,2,38], suggest that a major reduction of the solution space of metabolic models to a more biologically meaningful subspace is a general property of ecModels. However, flux variability is an intrinsic characteristic of metabolism; therefore, metabolic models with highly constrained solution spaces may exclude some biological capabilities of organisms, which are not compatible with the set of constraints used for the analysis (exchange fluxes, growth rates and even profiles of kinetic parameters, considered as condition-independent in ecModels).

Here, the predictive capabilities of ec*i*ML1515 and *i*JL1678 ME-model (both for *E. coli*) for cellular growth and global protein demands on diverse environments were compared. The major improvement in predicted maximum growth rates, together with a comparable performance on quantification of protein demands, shown by ec*i*ML1515 suggest that, despite its mathematical and conceptual simplicity, the GECKO formalism is a suitable framework for quantitative probing of metabolic capabilities, compatible with the widely used FBA method and without the need of excessive complexity or computational power. Nevertheless, ME-models provide a much wider range of predictions that explore additional processes in cell physiology with great detail. Direct comparison between the predictions of these modeling formalisms, suggest that ME-models performance can be improved by incorporation of either curated or systematically retrieved kinetic parameters that are suitable for the modeled organisms.

Simpler modeling frameworks that account for protein or enzyme constraints in metabolism, such as flux balance analysis with molecular crowding (FBAwMC)[16,17], metabolic modeling with enzyme kinetics (MOMENT)[23], and constrained allocation flux balance analysis (CAFBA)[21], have also been developed and used to explore microbial cellular growth[16,17,21] and overflow metabolism[16,23]. These methods have overcome the lack of reported parameters for some specific reactions either by incorporation of proteomics measurements and prior flux distributions[23], manual curation and sampling procedures[16,17] or even by lumping protein demands by functionally related proteome groups. In contrast, the new version of the GECKO toolbox provides a systematic and robust parameterization procedure, leveraging the vastly accumulated knowledge of biochemistry research stored in public databases, ensuring the incorporation of biologically meaningful kinetic parameters even for poorly studied reactions and organisms.

The applicability of these other simple modeling formalisms to models for diverse species is limited as none of these methods has been provided as part of a generalized model-agnostic software implementation. Recently, a simplified variant of the MOMENT method (sMOMENT) was developed and embedded into an automated pipeline for generation and calibration of enzyme-constrained models of metabolism (AutoPACMEN)[70]. The pipeline was tested on the generation of an enzyme-constrained version of the *i*JO1366 metabolic reconstruction for *E. coli*, which also showed consistency with experimental data. This work represented a step forward in the field of constrain-based metabolic modeling, as it contributed to standardization of model generation and facilitating their utilization and applicability to other cases. However, due to the intrinsic trade-off between model simplicity and descriptive representation, a limitation of the sMOMENT method is its simplification of redundancies in metabolism, which just accounts for the

optimal way of catalyzing a given biochemical reaction, discarding the representation of alternative isoforms that might be relevant under certain conditions. In GECKO ecModels, all enzymes for which a gene-E.C. number relationship exists are included in the model structure. As traditional FBA simulations rely on optimality principles one could, in principle, expect the same predicted flux distributions by sMOMENT and GECKO ecModels. Nonetheless, the explicit incorporation of all enzymes in a metabolic network enables explanation of protein expression profiles that deviate from optimality in order to gain robustness to changes in the environment, as it has been recently shown by the integration of a regulatory nutrient-signaling Boolean network together with an ecModel for *S. cerevisiae*'s central carbon metabolism[71].

In conclusion, GECKO 2.0 together with the development of the automated pipeline *ecModels container* facilitates the generation, standardization, utilization, exchange and community development of ecModels through a transparent version-controlled environment. This tool provides a dynamic, and potentially increasing, catalog of updated ecModels trying to close the gap between model developers and final users and reduce the time-consuming tasks of model maintenance. We are confident that this will enable wide use of ecModels in basic science for obtaining novel insight into the function of metabolism, as well as in synthetic biology and metabolic engineering for design of strains with improved functionalities, e.g., for high-level production of valuable chemicals.

## Methods

**Automation pipeline and version-controlled hosting of the ecModels container.** The ecModels repository is used to version-control the pipeline code and the resulting models. The pipeline is restricted to 2 short Python files, whose role is to decide when models need to be updated based on a configuration file config.ini, and to consequently invoke the use of GECKO for each model. Updates are deemed necessary when either the underlying dependencies (i.e., GECKO, RAVEN and COBRA toolboxes, the Gurobi solver, and libSMBL) or the source GEMs are independently updated to a new version (release) in their respective repositories.

The pipeline is designed be automatic and to not require supervision. It was developed to work with both version-controlled GEMs and GEMs downloadable from a URL, updating the version in the configuration after a new ecModel is obtained. For easy review, the pipeline log is publicly available under the *Actions* tab of the GitHub repository. The computation is performed through a self-hosted GitHub runner, further leveraging the transparent nature of the GitHub platform and the *git* version-control system. The resulting ecModel and updated configuration are committed to the repository, with the changes being made available for review through a pull request. Additionally, the GECKO output is also replicated in the pull request body. The *ecModels container* thus continues the transparency and reproducibility of the source models.

**Quantification of absolute protein concentrations for *S. cerevisiae, Y. lipolytica* and *K. marxianus*.** Total protein extraction for the strains *Saccharomyces cerevisiae* CEN.PK113-7D (standard, low pH, high temperature, osmotic stress), *Kluyveromyces marxianus* CBS6556 (standard, low pH, high temperature, osmotic stress) and *Yarrowia lipolytica* W29 (standard, low pH, high temperature) was conducted as described in the supplementary methods section in the Supplementary Information File 1. Three reference samples (hereafter, 'bulk' samples), one per strain, were constructed by pooling 5 µg of each experimental sample. Aliquots of 15 µg of total protein extract from each sample (3 strains x 4 conditions x 3 replicates) and the three bulks were separated on one-dimensional sodium dodecyl-sulfate–polyacrylamide gel electrophoresis short-migration gels (1 × 1 cm lanes, Invitrogen, NP321BOX). Yeast proteins digestion was performed on excised bands from gel gradient and digested peptides of UPS2 (Sigma) were used as external standards for absolute protein quantification (more details in the supplementary methods section in the Supplementary Information File 1). Four microliters of the different peptide mixtures (800 ng for yeast peptides and 949 ng for bulks) were analyzed using an Orbitrap Fusion™ Lumos™ Tribrid™ mass spectrometer (Thermo Fisher Scientific).

Protein identification was performed using the open-source search engine X! Tandem pipeline 3.4.4[72]. Data filtering was set to peptide *E*-value < 0.01 and protein log(*E*-value) < −3. Relative quantification of protein abundances was carried out using the Normalized Spectral Abundance Factor (NSAF)[73] and the NSAF values obtained from UPS2 proteins in bulk samples were used to determine the suitable regression curves that allowed the conversion from relative protein abundance into absolute terms. The regression curves parameters for protein abundance quantification are shown in the supplementary methods section in the Supplementary Information File 1.

**Simulation of condition-dependent flux distributions.** Simulation of cellular phenotypes for conditions of environmental stress at low dilution rates with GEMs were performed by first setting bounds on measured glucose uptake and byproduct secretion rates according to experimental data from previous studies on chemostats[64]. Then the biomass production rate was constrained (both upper and lower bounds) with the experimental dilution rate (0.1 h$^{-1}$). Maximization of the non-growth associated maintenance pseudo-reaction was set as an objective function for the parsimonious FBA problem as a representation of the additional energy demands for regulation of cellular growth at non-optimal conditions. The same procedure was followed for simulations with ecModels constrained by a total protein pool. For the case of ecModels with proteomics constraints, the same set of constraints was used but the objective function was set as minimization of the total usage of unmeasured proteins, assuming that the regulatory machinery for stress tolerance is represented by the condition-specific protein expression profile.

**Prediction of microbial batch growth rates.** Batch cellular growth was simulated by allowing unconstrained uptake of all nutrients present in minimal mineral media, enabling a specific carbon source uptake reaction for each case while blocking the rest of the uptake reactions and allowing unconstrained secretion rates for all exchangeable metabolites. Maximization of the biomass production rate was used as an objective function for the resulting FBA problem. For prediction of total protein demands on unlimited nutrient conditions, media constraints were set as expressed above and experimental batch growth rate values were fixed as both lower and upper bounds for the biomass production pseudo-reaction. The total protein pool exchange pseudo-reaction was then unconstrained and set as an objective function to minimize, assuming that when exposed to unlimited availability of nutrients the total mass of protein available for catalyzing metabolic reactions becomes the limiting resource for cells. The solveLP function, available in the RAVEN toolbox (v2.4.3), was used for solving all FBA problems in this study.

**Reporting summary.** Further information on research design is available in the Nature Research Reporting Summary linked to this article.

## Data availability

Mass spectrometry raw data that support the findings of this study have been deposited in PRIDE database[74] with the dataset identifier PXD012836. The processed proteomics datasets are available in our GitHub repository at: https://github.com/SysBioChalmers/GECKO2_simulations/tree/v1.0.1/data/proteomics. All collected kinetic data for the study presented in Supplementary Information File are available at: https://github.com/SysBioChalmers/Enzyme-parameters-analysis/tree/master/data. The generated computational models used for this study are available at: https://github.com/SysBioChalmers/ecModels/tree/v1.0.0. Data for reproduction of all main and supplementary figures are provided in the Sournce Data: Data Source file 1, Data Source File 2, and Data Source File 3. Source data are provided with this paper.

## Code availability

The source code of the updated GECKO toolbox is available at: https://github.com/SysBioChalmers/GECKO/releases/tag/v2.0.2[75]. The source code for ecModels container can be accessed at: https://github.com/SysBioChalmers/ecModels/tree/v1.0.0[76]. All custom scripts for simulations included in this study can be found at: https://github.com/SysBioChalmers/GECKO2_simulations/releases/tag/v1.0.1[77]. All the necessary scripts for reproducing the $k_{cat}$ parameters analysis in the Supplementary Information File 1 are available at: https://github.com/SysBioChalmers/Enzyme-parameters-analysis/releases/tag/v1.0.0[78]. All of these repositories are public and open to collaborative continuous development.

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

## Acknowledgements

We are grateful to Feiran Li, Raphaël Ferreira, Jonathan Robinson, and all the GECKO users that have provided feedback for improving our toolbox and extending its range of applications and to the CHASSY project consortium for having motivated and supported this work. This project has received funding from the European Union's Horizon 2020 Framework Programme for Research and Innovation—Grant Agreements No. 720824 to I.D., A.M.O., C.H., V.S., and J.P.M. 686070 to B.S. and 760798 to M.A. This work was also supported by the Knut and Alice Wallenberg Foundation and The Novo Nordisk Foundation—Grant no. NNF10CC1016517 to J.N.

## Author contributions

Conceptualization: I.D., B.S., M.A., E.J.K., and J.N.; data curation: A.M.O., and C.H.; formal analysis: I.D.; funding acquisition: J.N.; methodology: I.D.; project administration: J.N.; software: I.D., B.S., and M.A.; supervision: V.S., J.P.M., N.S., and J.N.; validation: I.D.; visualization: I.D.; writing—original draft: I.D., B.S., M.A, E.J.K.; writing—review and editing: I.D., M.A., V.S., J.P.M., N.S., and J.N.

## Funding

## Competing interests

The authors declare no competing interests.
